

# HIF inhibitor topotecan has a neuroprotective effect in a murine retinal ischemia-reperfusion model

Hiromitsu Kunimi[1,2], Yukihiro Miwa[1,2], Yusaku Katada[1,2], Kazuo Tsubota[1] and Toshihide Kurihara[1,2]

[1] Department of Ophthalmology, School of Medicine, Keio University, Tokyo, Japan
[2] Laboratory of Photobiology, School of Medicine, Keio University, Tokyo, Japan

## ABSTRACT

**Purpose**. The therapeutic approach for retinal ganglion cell (RGC) degeneration has not been fully established. Recently, it has been reported that hypoxia-inducible factor (HIF) may be involved with retinal neurodegeneration. In this study, we investigated neuroprotective effects of a HIF inhibitor against RGC degeneration induced in a murine model of retinal ischemia-reperfusion (I/R).

**Methods**. Eight-weeks-old male C57/BL6J mice were treated with intraperitoneal injection of a HIF inhibitor topotecan (1.25 mg/kg) for 14 days followed by a retinal I/R procedure. Seven days after the I/R injury, the therapeutic effect was evaluated histologically and electrophysiologically.

**Results**. The increase of HIF-1α expression and the decrease of retinal thickness and RGC number in I/R were significantly suppressed by administration of topotecan. Impaired visual function in I/R was improved by topotecan evaluated with electroretinogram and visual evoked potentials.

**Conclusions**. Topotecan administration suppressed HIF-1a expression and improved RGC survival resulting in a functional protection against retinal I/R. These data indicated that the HIF inhibitor topotecan may have therapeutic potentials for RGC degeneration induced with retinal ischemia or high intraocular pressure.

Corresponding authors
Kazuo Tsubota, tsubota@z3.keio.jp
Toshihide Kurihara, kurihara@z8.keio.jp

## INTRODUCTION

Tissue ischemia and hypoxia may induce irreversible neuronal degeneration. In the eye, retinal ganglion cell (RGC) degeneration is observed accompanied with ischemia in central retinal artery occlusion and ischemic optic neuropathy. Recently, optical coherence tomography (OCT) angiography technology reveals that capillary dropout is correlated with the decreased RGC layer thickness and visual field defects in glaucoma (*Takusagawa et al., 2017*). Mounting evidence suggests that retinal ischemia plays an important role in RGC degeneration (*Wang et al., 2002*). These damages directly affect visual acuity and visual field; however, therapeutic options for RGC degeneration are limited and establishment of a protective approach for RGC is desired.

Hypoxia-inducible factor (HIF) is a transcriptional factor that has a pivotal role in cellular adaptive response to hypoxic condition. Stabilization and activation of HIF induces cell survival under hypoxia including neovascularization, cell respiration, apoptosis, glucose metabolism, and embryogenesis (*Semenza, 2011*). In the retina, HIF plays a critical role in development, physiology and pathology related with angiogenesis and anaerobic metabolism (*Kurihara et al., 2016*; *Kurihara, 2018*). It has also reported that HIF-1α expression is increased in human glaucomatous retina (*Tezel & Wax, 2004*) suggesting a correlation between chronic RGC degeneration and HIF activation.

Retinal ischemia-reperfusion (I/R) is a well-established animal model to induce RGC degeneration (*Sellés-Navarro et al., 1996*; *Vidal-Sanz et al., 2001*; *Hartsock et al., 2016*; *Liu et al., 2019*). This model mimics an acute retinal artery occlusion such as CRAO or RGC death such as glaucoma. A previous report showed a neuroprotective effect of carnosine against I/R with decrease of HIF-1α expression in the retina (*Ji et al., 2014*). However, the relation between retinal neurodegeneration and HIF remains unclear. Topotecan is a topoisomerase inhibitor and is also known as a potent HIF inhibitor (*Rapisarda et al., 2002*). Recently, we reported that topotecan prevented retinal neovascularization and impaired visual function in a murine model of oxygen-induced retinopathy (*Miwa et al., 2019*), while it remains unclear that pharmacological HIF inhibition is effective for RGC degeneration. In this study, we investigated the protective effect of topotecan for RGC in a murine model of retinal I/R.

## MATERIALS & METHODS

### Ethics of animal research

All procedures for animal experiments were approved by IACUC of Keio University (Approval Number 2808), and were in accordance with NIH guidelines for work with animals, ARVO statement for the Use of Animals in Ophthalmic and Vision Research and ARRIVE guidelines.

### Drug administration

All experiments were carried out with 8-weeks-old male C57/BL6J mice (CLEA Japan, Japan). Animals were divided into two groups and intraperitoneally injected with phosphate buffered saline (PBS) or Topotecan dissolved in PBS (1.25 mg/kg, #14129, Cayman Chemical, United States) once per day for 14days prior to the retinal I/R. All mice were maintained on a standard rodent diet (MF, Oriental Yeast Co., Ltd, Japan) and given free access to water. All cages were maintained under controlled lighting (12 h light/12 h dark).

### Murine retinal I/R experiment

The murine model of retinal I/R followed with Western blotting, qPCR, retinal thickness evaluation, RGC retrograde labeling and electrophysiological evaluation were performed as previously described (*Kunimi et al., 2019*). Specifically, we performed Western blotting using mouse anti-HIF-1α (1:1500; #36169; CST, Danvers, MA, USA) and mouse anti-β-actin (1:4000; #A5316; Sigma-Aldrich, St. Louis, MO, USA) primary antibodies. The primers for qPCR were synthesized by Thermo Fisher Scientific, Waltham, MA, USA.
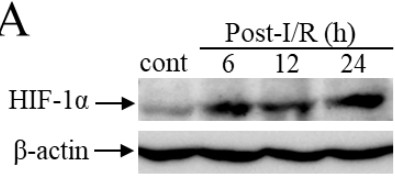
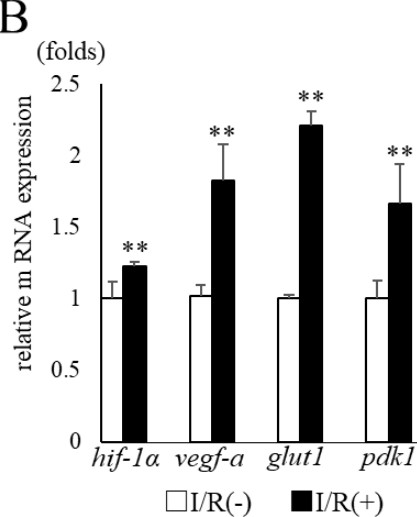

**A** Post-I/R (h)
cont 6 12 24
HIF-1α →
β-actin →

**B** (folds)
relative m RNA expression

hif-1α vegf-a glut1 pdk1
□ I/R(-) ■ I/R(+)

**Figure 1** **HIF-1α and its target genesexpression in post- I/R retina.** (A) Western blots show retinal HIF-1α expression is increased and maintained 6 h after I/R injury. (B) *Hif-1 α* and its representative target genes were upregulated in post-I/R retina detected by qPCR ($n = 5$). *Gapdh* was used as the internal control. Error bars indicate the standard error. Cont; control. $**p < 0.01$, Mann-Whitney's $U$ test.

## Statistical analysis

The data were presented as the mean $\pm$ SD. Comparison of two experimental conditions was evaluated using Mann–Whitney's $U$-test. A $p < 0.05$ was considered statistically significant.

## RESULTS

### Expression of HIF-1α and its target genes after retinal I/R injury

In the current study, we examined murine retinal I/R model to induce RGC degeneration. HIF-1α protein expression in the retina was increased in 6 h after I/R injury (Fig. 1A) with significant upregulation of *hif-1 α* and its representative target genes (*vegf-a, glut1, pdk1*) (*hif-1 α*: $p = 0.009$, *vegf-a*: $p = 0.009$, *glut1*: $p = 0.009$, *pdk1*:$p = 0.009$, respectively) (Fig. 1B). These data indicated that retinal HIF-1α signaling was activated with I/R injury.

### Change of HIF-1α and target gene expressions with topotecan administration

Next, we administered topotecan intraperitoneally in order to inhibit HIF-1α pharmacologically in mice. The increased HIF-1α protein expression in post-I/R retinas ($p = 0.009$) was significantly ($p = 0.009$) suppressed in topotecan-treated mice compared to controls (Figs. 2A, 2B). The upregulated retinal *hif-1α* and the target genes were also significantly suppressed except for *pdk1* in treated mice compared to controls (*hif-1α*: $p = 0.009$, *vegf-a*: $p = 0.016$, *glut1*: $p = 0.009$, *pdk1*: $p = 0.028$, respectively) (Fig. 2C). These results suggested that systemic administration of topotecan inhibited increased HIF-1α and upregulated target gene expression in post I/R retinas.

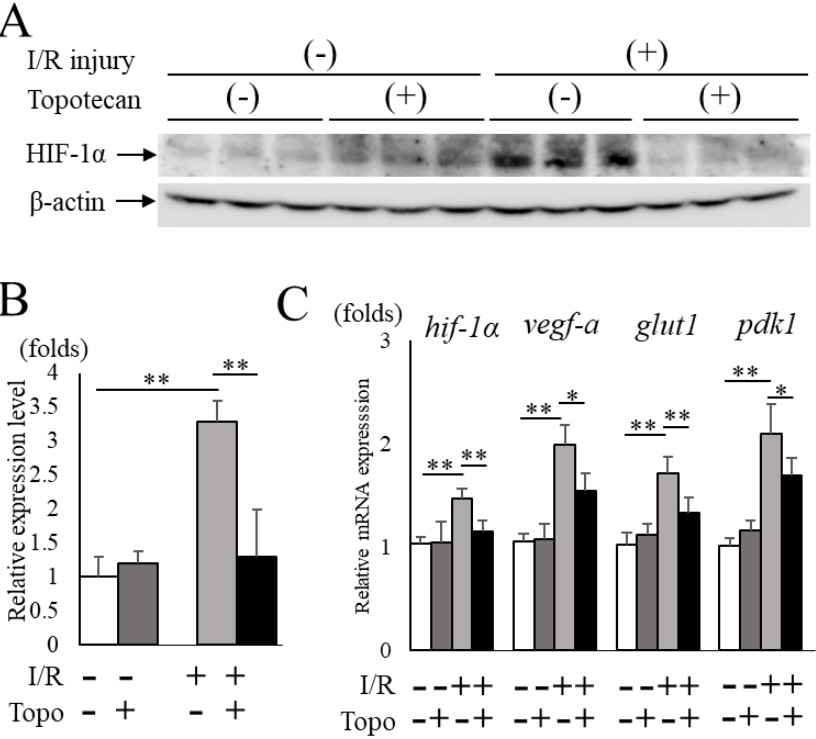

**Figure 2** **Topotecan administration suppresses increased HIF-1α and upregulated targetgenes in I/R retinas.** (A) Western blots for HIF-1α and $\beta$-actin in control or I/R retinas with or without topotecan administration ($n = 5$). (B) Quantification of the blots indicating that topotecan administration suppressed increased HIF-1α expression. (C) *Hif-1α* and its representative target genes detected by qPCR ($n = 5$). Note that upregulated genes were suppressed by topotecan administration. *Gapdh* was used as the internal control. Error bars indicate the standard error. *$p < 0.05$, **$p < 0.01$, Mann–Whitney's $U$ test.

## Improvement of RGC survival with topotecan administration in post-I/R retinas

We examined the retinal thickness to evaluate the effect of topotecan morphologically with OCT. Total retinal thickness was significantly ($p = 0.021$) thinner in a week after I/R injury, while topotecan group showed significantly ($p = 0.021$) thicker retina compared to control (Fig. 3). We further examined fluorogold retrograde labeling of RGCs to assess the cell survival 7 day after I/R injury. While the number of RGCs were significantly ($p = 0.009$) decreased in post-I/R retinas, topotecan administration significantly ($p = 0.009$) suppressed the decrease of RGC number (Fig. 4). These results indicated that topotecan administration had a neuroprotective effect improving RGC survival against retinal I/R damage.

## Protective effect of topotecan for the impaired visual function with I/R injury

To evaluate the change of retinal function with topotecan treatment, we examined ERG after I/R injury. In this study, ERG waveforms in three different stimulating conditions were recorded 7 days after I/R injury. The amplitudes was significantly decreased in each

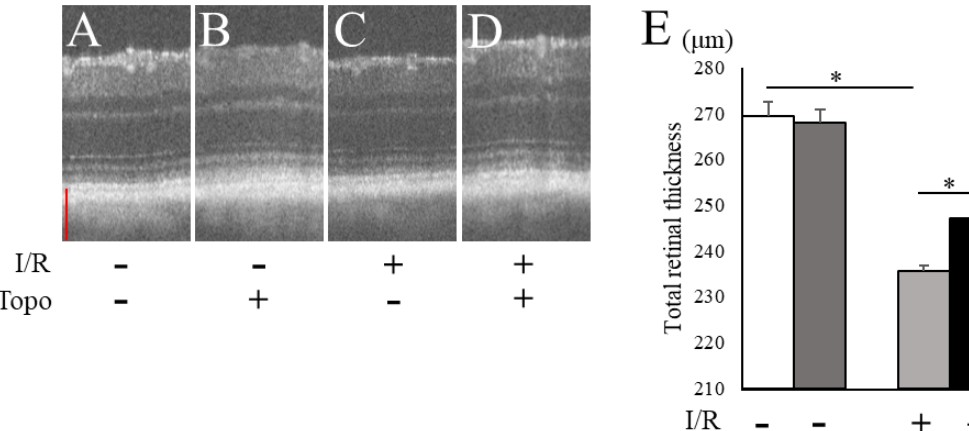

**Figure 3** **Evaluation of totalretinal thickness with OCT.** (A–D) Representative OCT images from each group. Scale bar; 100 μm. (E) The average of total retinal thickness quantified in OCT ($n = 4$). Note that decrease of retinal thickness was suppressed by topotecan administration post-I/R injury. Error bars indicate the standard error. $*p < 0.05$, Mann–Whitney's $U$ test.

condition after I/R injury (rod b-wave: $p = 0.009$, mix a-wave: $p = 0.009$, mix b-wave: $p = 0.009$, cone b-wave: $p = 0.009$, respectively), while topotecan administration suppressed the decrease of amplitudes with I/R injury except for cone b-wave (rod b-wave: $p = 0.028$, mix a-wave: $p = 0.016$, mix b-wave: $p = 0.006$, cone b-wave: $p = 0.056$, respectively) (Fig. 5). In addition to ERG, we also assessed VEP to evaluate the protective effect of topotecan in I/R injury. I/R injured mice showed a significant ($p = 0.009$) decrease of amplitudes and a significantly ($p = 0.009$) prolonged implicit time. On the other hand, the decrease of VEP amplitudes was significantly ($p < 0.016$) suppressed with topotecan administration (Fig. 6). These results suggested that topotecan had a neuroprotective effect against I/R damage functionally.

## DISCUSSION

In this study, we focused on the role of HIF-1α in the RGC degeneration. The contribution of HIF to RGC death or nerve fiber degeneration has not well been documented previously. In the current study, RGC loss was observed after I/R injury accompanied with retinal excessive expression of HIF-1α while HIF-1α inhibition with topotecan protected RGC morphologically and functionally in I/R injured retinas. To our best knowledge, this is the first study to show a protective effect of HIF inhibitor to RGC degeneration. In the current study, we injected topotecan before the I/R procedure to obtain the maximum effect of the drug at the onset of the model. Administration of the drug after I/R is required to prove the therapeutic concept for diseases in future studies.

To date, several neuroprotective materials for RGC degeneration have been reported. Synthetic steroid showed RGC protection via suppressing the microglial inflammation (*Sun et al., 2019*). Rapamycin, an antibiotic agent, promoted autophagy in the retina and improved RGC survival (*Russo et al., 2018*). Other antibacterial drug, minocycline,

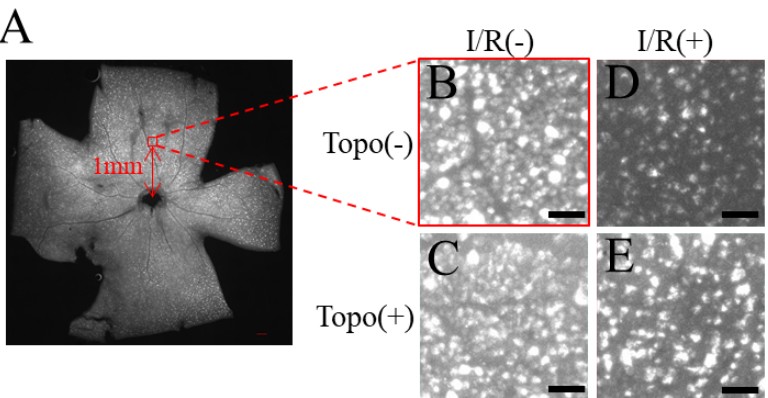

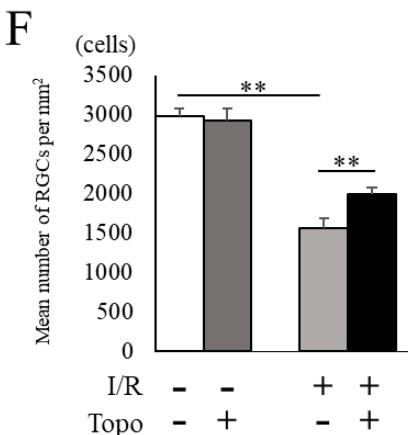

**Figure 4 Fluorogold retrograde labeling of RGCs.** (A) A representative quadrant retina with fluorogold-labeled RGCs. 200 µm square with red at one mm from optic disc head indicates the area for RGC densitometry. (B–E) Magnified images for control and post-I/R retina with or without topotecan administration. Scale bars; 200 µm in quadrant retina, 50 µm in magnified images. (F) The quantification of RGC density for each group ($n = 5$). Note that decrease of RGCs was suppressed by topotecan administration. Error bars indicate the standard error. ** $p < 0.01$, Mann–Whitney's $U$ test.

had a neuroprotective effect in retinal I/R (*Huang et al., 2018*). Bevacizumab, a human monoclonal antibody to VEGF which is downstream of HIF-1α, also reduced RGC apoptosis in a rat model of I/R (*Kohen et al., 2018*). In addition, some of dietary factors also showed retinal neuroprotective effects. Resveratrol enhanced the survival of RGCs against I/R via anti-inflammatory action (*Luo et al., 2018*). Xue-Fu-Zhu-Yu, one of traditional Chinese medicine protected RGC from retinal ischemia damage (*Tan et al., 2017*). Thus, these substrates showing RGC protective effects with various mechanisms are expected for clinical application.

There are some studies conducted to investigate pathways of RGC apoptosis. Caspases had been explored to play an important role in neuronal cell apoptosis of the inner retinal layer in the early stage of I/R (*Lam, Abler & Tso, 1999*), while RGC necrosis was induced via extracellular signal-regulated kinase 1/2-receptor-interacting protein kinase 3 pathway
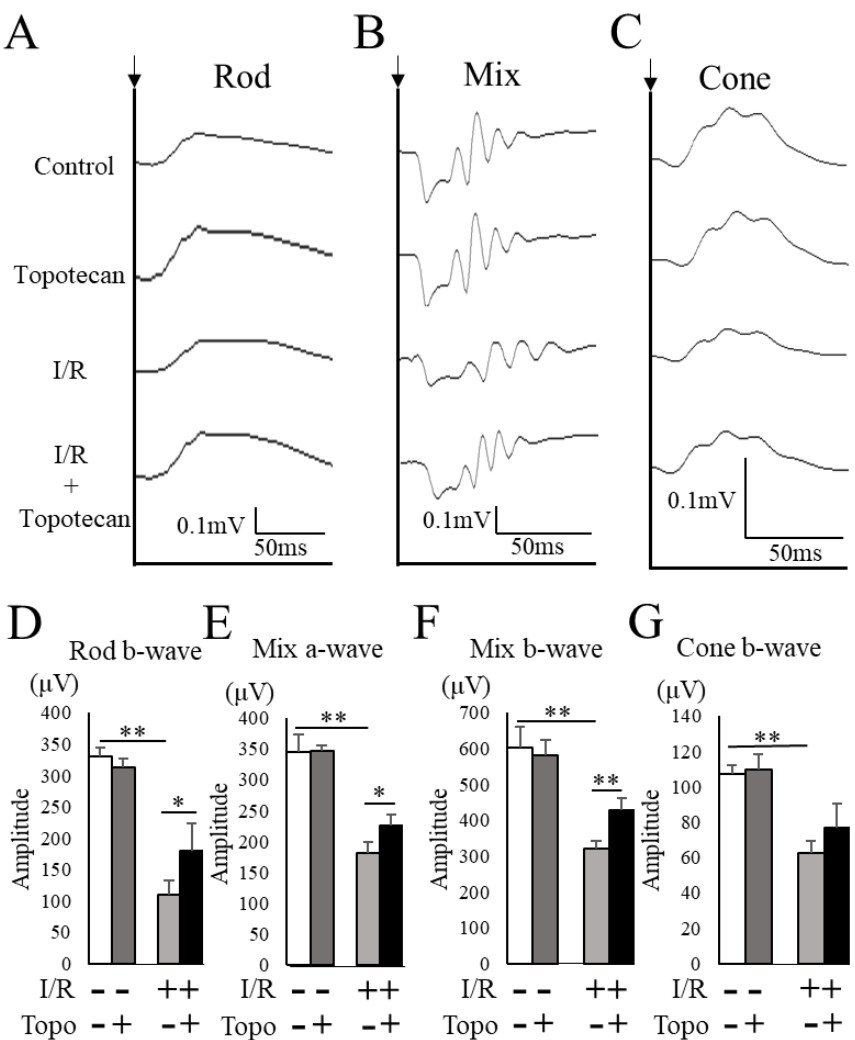

**Figure 5 Retinal functionevaluated with ERG.** (A–C) Representative ERG waveforms for rod, mixed, and cone conditions. Black arrows indicate the timing of the stimulation. The averaged amplitudes were shown for rod b-wave (D), mixed a-wave (E), mixed b-wave (F), and cone b-wave (G) ($n = 5–6$). Note that most of decreased amplitudes were suppressed by topotecan administration. Error bars indicate the standard error. $*p < 0.05$, $**p < 0.01$, Mann–Whitney's $U$ test.

(*Gao, Andreeva & Cooper, 2014*). Ca-phospholipid binding protein annexin A1 is shown to increase IL-1β expression promoting RGC death via p65 pathway (*Zhao et al., 2017*).

As previously reported, the HIF-1α expression in each cell type is functionally related to the phenotype of angiogenesis physiologically and pathologically. Critical roles of HIF-1α in retinal neuron including RGC (*Nakamura-Ishizu et al., 2012*), muller cells (*Lin et al., 2011*), astrocytes, and microglia (*Kurihara et al., 2011*) have been shown by utilizing cell type specific conditional gene knockout technology. Although the specific cell type contributing to the effect of the drug administration has not been identified at this point, we will explore the contribution of HIF in each cell type in future studies. It has been reported that the activation of NRF2/HO-1 pathway (a downstream of HIF-1α) in RGCs

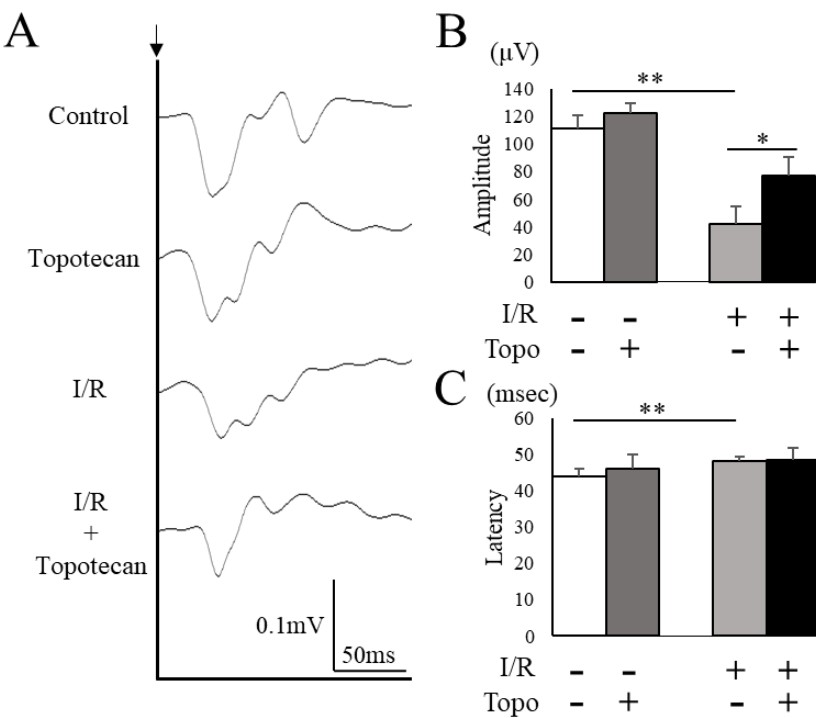

**Figure 6  Evaluation of visual function detected with VEP.** (A) Representative VEP waveforms from control and post-I/R retina with or without topotecan administration. A black arrow indicates the timing of the stimulation. (B) The average of VEP amplitudes ($n = 5$). Note that decrease of VEP amplitude was suppressed by topotecan administration. (C) The average of VEP implicit time. Error bars indicate the standard error. *$p < 0.05$, **$p < 0.01$, Mann–Whitney's $U$ test.

inhibited RGC degeneration in a rat retinal I/R model (*Varga et al., 2013*; *He et al., 2014*). In a rat chronic high IOP model, HIF-1α expression was increased in muller cells and astrocytes but not in microglia (*Ergorul et al., 2010*). These results indicated that the *in vivo* phenotype might not be mediated by a single cell type but by the sum of these compensates.

Previously, we described that hypoxic retinal pigment epithelium induced degeneration of photoreceptor cells altering glucose and lipid metabolism through HIF expression (*Kurihara et al., 2016*). In contrast, it has been reported that the expression of HIF downstream genes such as *Erythropoietin* (*Grimm et al., 2002*; *Sullivan, Kodali & Rex, 2011*) or *Vegf* (*Foxton et al., 2013*) have a neuroprotective effect against neuronal damages in the retina. Thus, further studies are needed to conclude the contribution of HIF in the degenerative retina. HIF may be related to the neurodegeneration in a cell autonomous manner such as apoptosis (*Greijer & Van der Wall, 2004*); besides, non-cell autonomous mechanisms including recruitment of cytotoxic inflammatory cells or activation of supporting cells can be considered to induce RGC damages (*Zera & Zastre, 2017*). HIF inhibition may suppress these negative reactions against the RGC survival in the process of the degeneration. Since HIF is a transcription factor, further study should be required to elucidate the downstream mechanism of HIF in the RGC degeneration. Taken together, a HIF inhibitor topotecan had a neuroprotective effect for RGCs in retinal ischemia

following hyper intraocular pressure. This study suggested that pharmacological HIF inhibition may be possible candidates for RGC degeneration induced with retinal ischemia or high intraocular pressure.

## CONCLUSIONS

In the current study, we hypothesized that HIF was involved with RGC degeneration in acute retinal I/R. This is the first research to indicate that topotecan had a protective effect against retinal I/R histologically and electrophysiologically suppressing elevated HIF-1α expression. Although further study is needed to show the specificity of HIF in retinal neurodegeneration, topotecan is a possible drug to protect RGC from retinal ischemia and high intraocular pressure.

### Abbreviations

| | |
|---|---|
| **IACUC** | Institutional Animal Care and Use Committee |
| **NIH** | National Institutes of Health |
| **ARVO** | Association for Research in Vision and Ophthalmology |
| **ARRIVE** | Animal Research: Reporting of In Vivo Experiments |

## ACKNOWLEDGEMENTS

We thank H Torii; S Ikeda; Y Hagiwara; K Mori; E Yotsukura; X Jiang; M Ibuki; C Shoda; N Ozawa; A Ishida; K Takahashi and K Kurosaki for critical discussions.

### Funding

This work was supported by Grants-in-Aid for Scientific Research (KAKENHI, number 15K10881 and 18K09424) from the Ministry of Education, Culture, Sports, Science and Technology (MEXT) to Toshihide Kurihara. The funders had no role in study design, data collection and analysis, decision to publish, or preparation of the manuscript.

### Grant Disclosures

The following grant information was disclosed by the authors:
Grants-in-Aid for Scientific Research: 15K10881, 18K09424.
Ministry of Education, Culture, Sports, Science and Technology.

### Competing Interests

The authors declare there are no competing interests.

### Author Contributions

- Hiromitsu Kunimi conceived and designed the experiments, performed the experiments, analyzed the data, contributed reagents/materials/analysis tools, prepared figures and/or tables, authored or reviewed drafts of the paper, approved the final draft.

- Yukihiro Miwa conceived and designed the experiments, contributed reagents/materials/analysis tools, prepared figures and/or tables, authored or reviewed drafts of the paper, approved the final draft.
- Yusaku Katada conceived and designed the experiments, contributed reagents/materials/analysis tools, authored or reviewed drafts of the paper, approved the final draft.
- Kazuo Tsubota conceived and designed the experiments, authored or reviewed drafts of the paper, approved the final draft.
- Toshihide Kurihara conceived and designed the experiments, analyzed the data, contributed reagents/materials/analysis tools, prepared figures and/or tables, authored or reviewed drafts of the paper, approved the final draft.

## Animal Ethics

The following information was supplied relating to ethical approvals (i.e., approving body and any reference numbers):

The Institutional Animal Care and Use Committee of Keio University provided full approval for this research (#2808).

## Data Availability

The raw data showing the uncropped images of western blots is available in Fig. S1.

## Supplemental Information

Supplemental information for this article can be found online at http://dx.doi.org/10.7717/peerj.7849#supplemental-information.

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
