# Peer review of "HIF inhibitor topotecan has a neuroprotective effect in a murine retinal ischemia-reperfusion model"

_PeerJ, doi:10.7717/peerj.7849_

## Round 0.1 · original submission · Major Revisions

I recommend that you address the comments of the reviewers, particularly in relation to experimental design and validity of the findings.

Reviewer 1 ·

Basic reporting

Overall, the writing is clear and unambiguous. However, some cosmetic changes to the English writing would improve the readability of the manuscript.

The background is sufficient, and references are pertinent. Overall the basic reporting is good.

The top right panel in Figure 4A (I/R(+)) is too dark. Please adjust the brightness to match the others (unless the authors have a good reason not to).

Experimental design

The research question is well defined, interesting, and timely. It is true that neuroprotection of RGCs in various retinal diseases is an unmet need in the field. The model employed is relevant, and the data are clean and presented clearly. Methods are written clearly.

Validity of the findings

Data are robust, sound, and controlled. Speculation is reasonable.

Additional comments

The authors show data that preventing HIF stabilization provides functional/anatomical rescue in I/R mice.
This is a simple, straightforward study and the findings are clear. The topic is timely, and interesting.

I have some specific comments:
1. Can the authors comment on the revelance of the findings to human disease. i.e. Relevance of animal model/contribution of ischemia to RGC degeneration in retinal disease.
2. Can the authors comment more about what is known about HIF1a function in RGCs and supporting cells (astrocytes/microglia). Can the authors rule out an involvement of topetecan in astrocytes/microglia/mueller cells?
3. The authors should clarify that HIF1a is active in multiple other retinal cell-types. Can the authors comment if there may be off target effects?

Reviewer 2 ·

Basic reporting

The article evaluated the neuroprotective role of topotecan, a HIF inhibitor. The language used is clear, not ambiguous. Introduction and background show context. All references used are relevant. The structure of the manuscript is in accordance with the PeerJ standard.
The article has an important contribution to ophthalmology. However, my recommendation is that the authors should clarify some points of the study before it is accepted for publication.

Experimental design

Why did the authors inject topotecan prior to the induction of retinal injury? As it is a possible therapeutic candidate, it would be interesting to evaluate the protective effect of topotecan after induction of the lesion.
Please define normal saline. Is it 10% NaCl?
Define IOP at first indication (intraocular pressure) - line 79
Which cocktail inhibitor protease was used in this study? Please enter the manufacturer's catalog number
Enter manufacturer's catalog number for all kits cited in the study
The methods were presented in the sequence in which they were performed? Rearrange the M&M section, if necessary.
Was the electrophysiological evaluation done according to the previous literature? Provide a reference, please.
Detail the preparation or acquisition of the primers.
Inform if the data were tested for statistical normality

Validity of the findings

The results were not discussed correctly. The authors did not relate their findings to previous literature. Please review this section carefully
The figures are adequate
Standardize the terminology. Please use RGC degeneration (not RGC neurodegeneration)
This reviewer understands that further studies are needed to elucidate the mechanisms of HIF as well as its inhibitory drugs. However, speculations would be welcome in the discussion.
Review the sentence - "This study suggested that pharmacological HIF inhibition may be candidates for RGC degeneration induced with renal ischemia or high intraocular pressure."

Additional comments

The article evaluated the neuroprotective role of topotecan, a HIF inhibitor. The language used is clear, not ambiguous. Introduction and background show context. All references used are relevant. The structure of the manuscript is in accordance with the PeerJ standard.
The article has an important contribution to ophthalmology. However, my recommendation is that the authors should clarify some points of the study before it is accepted for publication.
Why did the authors inject topotecan prior to the induction of retinal injury? As it is a possible therapeutic candidate, it would be interesting to evaluate the protective effect of topotecan after induction of the lesion.
Please define normal saline. Is it 10% NaCl?
Define IOP at first indication (intraocular pressure) - line 79
Which cocktail inhibitor protease was used in this study? Please enter the manufacturer's catalog number
Enter manufacturer's catalog number for all kits cited in the study
The methods were presented in the sequence in which they were performed? Rearrange the M&M section, if necessary.
Was the electrophysiological evaluation done according to the previous literature? Provide a reference, please.
Detail the preparation or acquisition of the primers.
Inform if the data were tested for statistical normality
The results were not discussed correctly. The authors did not relate their findings to previous literature. Please review this section carefully
The figures are adequate
Standardize the terminology. Please use RGC degeneration (not RGC neurodegeneration)
This reviewer understands that further studies are needed to elucidate the mechanisms of HIF as well as its inhibitory drugs. However, speculations would be welcome in the discussion.
Review the sentence - "This study suggested that pharmacological HIF inhibition may be candidates for RGC degeneration induced with renal ischemia or high intraocular pressure."

---

## Round 0.2 · Minor Revisions

The authors have substantially improved their manuscript and satisfactorily addressed all the reviewers' concerns.

However, there is a significant overlap in the text of this manuscript (Materials and Methods) with a recently published paper by the same authors (<https://www.ncbi.nlm.nih.gov/pubmed/31261724>).

We understand that the same methods are often reused, although there are only so many ways to describe a procedure. The homologous parts, therefore, should be rephrased.

Reviewer 1 ·

Basic reporting

The manuscript is well organized and is easy to read. Literature references are valid, and figures are clean and well described.

Experimental design

Research fills a knowledge gap, and is translatable to human disease. Methods are well described in this version.

Validity of the findings

Data are robust and conclusions are well stated.

Additional comments

The manuscript is much improved.

Reviewer 2 ·

Basic reporting

no comment

Experimental design

The study is well defined and relevant. Methods were described with sufficient detail.

Validity of the findings

Data are robust and conclusions well stated.

Additional comments

The authors followed all suggestions of this reviewer. The manuscript has significantly improved and all necessary information has been added.

---

## Round 0.3 · accepted · Accept

The authors have addressed all the Editor's and reviewers' requests.
The sentence at lines 82-84 should be rephrased before publication.